# Inverting Gradients - How easy is it to break privacy in federated learning?

**Jonas Geiping**[*]      **Hartmut Bauermeister** [*]      **Hannah Dröge** [*]

**Michael Moeller**

Dep. of Electrical Engineering and Computer Science
University of Siegen
{jonas.geiping, hartmut.bauermeister, hannah.droege,
michael.moeller }@uni-siegen.de

## Abstract

The idea of federated learning is to collaboratively train a neural network on a server. Each user receives the current weights of the network and in turns sends parameter updates (gradients) based on local data. This protocol has been designed not only to train neural networks data-efficiently, but also to provide privacy benefits for users, as their input data remains on device and only parameter gradients are shared. But how secure is sharing parameter gradients? Previous attacks have provided a false sense of security, by succeeding only in contrived settings - even for a single image. However, by exploiting a magnitude-invariant loss along with optimization strategies based on adversarial attacks, we show that is is actually possible to faithfully reconstruct images at high resolution from the knowledge of their parameter gradients, and demonstrate that such a break of privacy is possible even for trained deep networks. We analyze the effects of architecture as well as parameters on the difficulty of reconstructing an input image and prove that any input to a fully connected layer can be reconstructed analytically independent of the remaining architecture. Finally we discuss settings encountered in practice and show that even aggregating gradients over several iterations or several images does not guarantee the user's privacy in federated learning applications.

## 1 Introduction

Federated or collaborative learning [6, 28] is a distributed learning paradigm that has recently gained significant attention as both data requirements and privacy concerns in machine learning continue to rise [21, 14, 32]. The basic idea is to train a machine learning model, for example a neural network, by optimizing the parameters $\theta$ of the network using a loss function $\mathcal{L}$ and exemplary training data consisting of input images $x_i$ and corresponding labels $y_i$ in order to solve

$$\min_{\theta} \sum_{i=1}^{N} \mathcal{L}_{\theta}(x_i, y_i). \tag{1}$$

We consider a distributed setting in which a *server* wants to solve (1) with the help of multiple *users* that own training data $(x_i, y_i)$. The idea of federated learning is to only share the gradients $\nabla_{\theta}\mathcal{L}_{\theta}(x_i, y_i)$ instead of the original data $(x_i, y_i)$ with the server which it subsequently accumulates to

---

[*]Authors contributed equally.

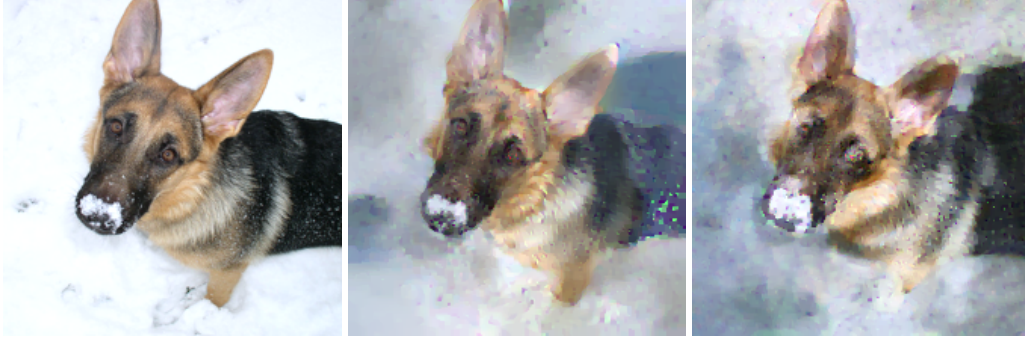

Figure 1: Reconstruction of an input image $x$ from the gradient $\nabla_\theta \mathcal{L}_\theta(x, y)$. Left: Image from the validation dataset. Middle: Reconstruction from a trained ResNet-18 trained on ImageNet. Right: Reconstruction from a trained ResNet-152. In both cases, the intended privacy of the image is broken. Note that previous attacks cannot recover either ImageNet-sized data [35] or attack trained models.

update the overall weights. Using gradient descent the server's updates could, for instance, constitute

$$\theta^{k+1} = \underbrace{\theta^k - \tau}_{\text{server}} \underbrace{\sum_{i=1}^{N} \nabla_\theta \mathcal{L}_{\theta^k}(x_i, y_i)}_{\text{users}}. \tag{2}$$

The updated parameters $\theta^{k+1}$ are sent back to the individual users. The procedure in eq. (2) is called *federated SGD*. In contrast, in *federated averaging* [17, 21] each user computes several gradient descent steps locally, and sends the updated parameters back to the server. Finally, information about $(x_i, y_i)$ can be further obscured, by only sharing the mean $\frac{1}{t} \sum_{i=1}^{t} \nabla_\theta \mathcal{L}_{\theta^k}(x_i, y_i)$ of the gradients of several local examples, which we refer to as the *multi-image* setting.

Distributed learning of this kind has been used in real-world applications where user privacy is crucial, e.g. for hospital data [13] or text predictions on mobile devices [3], and it has been stated that "Privacy is enhanced by the ephemeral and focused nature of the [Federated Learning] updates" [3]: model updates are considered to contain less information than the original data, and through aggregation of updates from multiple data points, original data is considered impossible to recover. In this work we show analytically as well as empirically, that parameter gradients still carry significant information about the supposedly private input data as we illustrate in Fig. 1. We conclude by showing that even *multi-image federated averaging* on realistic architectures does not guarantee the privacy of all user data, showing that out of a batch of 100 images, several are still recoverable.

**Threat model:** We investigate an *honest-but-curious* server with the goal of uncovering user data: The attacker is allowed to separately store and process updates transmitted by individual users, but may *not* interfere with the collaborative learning algorithm. The attacker may not modify the model architecture to better suit their attack, nor send malicious global parameters that do not represent the actually learned global model. The user is allowed to accumulate data locally in Sec. 6. We refer to the supp. material for further commentary and mention that the attack is near-trivial under weaker constraints on the attacker.

In this paper we discuss privacy limitations of federated learning first in an academic setting, honing in on the case of gradient inversion from one image and showing that

- Reconstruction of input data from gradient information is possible for realistic deep and non-smooth architectures with both, trained and untrained parameters.
- With the right attack, there is little "defense-in-depth" - deep networks are as vulnerable as shallow networks.
- We prove that the input to any fully connected layer can be reconstructed analytically independent of the remaining network architecture.

Then we consider the implications that the findings have for practical scenarios, finding that

- Reconstruction of multiple, separate input images from their averaged gradient is possible in practice, over multiple epochs, using local mini-batches, or even for a local gradient averaging of up to 100 images.

## 2 Related Work

Previous related works that investigate recovery from gradient information have been limited to shallow networks of less practical relevance. Recovery of image data from gradient information was first discussed in [25, 24] for neural networks, who prove that recovery is possible for a single neuron or linear layer. For convolutional architectures, [31] show that recovery of a single image is possible for a 4-layer CNN, albeit with a significantly large fully-connected (FC) layer. Their work first constructs a "representation" of the input image, that is then improved with a GAN. [35] extends this, showing for a 4-layer CNN (with a large FC layer, smooth sigmoid activations, no strides, uniformly random weights), that missing label information can also be jointly reconstructed. They further show that reconstruction of multiple images from their averaged gradients is indeed possible (for a maximum batch size of 8). [35] also discuss deeper architectures, but provide no tangible results. A follow-up [34] notes that label information can be computed analytically from the gradients of the last layer. These works make strong assumptions on the model architecture and model parameters that make reconstructions easier, but violate the threat model that we consider in this work and lead to less realistic scenarios.

The central recovery mechanism discussed in [31, 35, 34] is the optimization of an euclidean matching term. The cost function

$$\arg \min_x ||\nabla_\theta \mathcal{L}_\theta(x, y) - \nabla_\theta \mathcal{L}_\theta(x^*, y)||^2 \tag{3}$$

is minimized to recover the original input image $x^*$ from a transmitted gradient $\nabla_\theta \mathcal{L}_\theta(x^*, y)$. This optimization problem is solved by an L-BFGS solver [18]. Note that differentiating the gradient of $\mathcal{L}$ w.r.t to $x$ requires a second-order derivative of the considered parametrized function and L-BFGS needs to construct a third-order derivative approximation, which is challenging for neural networks with ReLU units for which higher-order derivatives are discontinuous.

A related, but easier problem, compared to the full reconstruction of input images, is the retrieval of input attributes [23, 10] from local updates, e.g. does a person that is recognized in a face recognition system wear a hat. Information even about attributes unrelated to the task at-hand can be recovered from deeper layers of a neural network, which can be recovered from local updates.

Our problem statement is furthermore related to model inversion [9], where training images are recovered from network parameters after training. This provides a natural limit case for our setting. Model inversion generally is challenging for deeper neural network architectures [33] if no additional information is given [9, 33]. Another closely related task is inversion from visual representations [8, 7, 20], where, given the output of some intermediate layer of a neural network, a plausible input image is reconstructed. This procedure can leak some information, e.g. general image composition, dominating colors - but, depending on the given layer it only reconstructs similar images - if the neural network is not explicitly chosen to be (mostly) invertible [11]. As we prove later, inversion from visual representations is strictly more difficult than recovery from gradient information.

## 3 Theoretical Analysis: Recovering Images from their Gradients

To understand the overall problem of breaking privacy in federated learning from a theoretical perspective, let us first analyze the question if data $x \in \mathbb{R}^n$ can be recovered from its gradient $\nabla_\theta \mathcal{L}_\theta(x, y) \in \mathbb{R}^p$ analytically.

Due to the different dimensionality of $x$ and $\nabla_\theta \mathcal{L}_\theta(x, y)$, reconstruction quality is surely is a question of the number of parameters $p$ versus input pixels $n$. If $p < n$, then reconstruction is at least as difficult as image recovery from incomplete data [4, 2], but even when $p > n$, which we would expect in most computer vision applications, the difficulty of regularized "inversion" of $\nabla_\theta \mathcal{L}_\theta$ relates to the non-linearity of the gradient operator as well as its conditioning.

Interestingly, fully-connected layers take a particular role in our problem: As we prove below, the input to a fully-connected layer can always be computed from the parameter gradients analytically independent of the layer's position in a neural network (provided that a technical condition, which

prevents zero-gradients, is met). In particular, the analytic reconstruction is independent of the specific types of layers that precede or succeed the fully connected layer, and a single input to a fully-connected network can always be reconstructed analytically without solving an optimization problem.The following statement is a generalization of Example 3 in [24] to the setting of arbitrary neural networks with arbitrary loss functions:

**Proposition 3.1.** *Consider a neural network containing a biased fully-connected layer preceded solely by (possibly unbiased) fully-connected layers. Furthermore assume for any of those fully-connected layers the derivative of the loss $\mathcal{L}$ w.r.t. to the layer's output contains at least one non-zero entry. Then the input to the network can be reconstructed uniquely from the network's gradients.*

*Proof.* In the following we give a sketch of the proof and refer to the supplementary material for a more detailed derivation. Consider an unbiased full-connected layer mapping the input $x_l$ to the output, after e.g. a ReLU nonlinearity: $x_{l+1} = \max\{A_l x_l, 0\}$ for a matrix $A_l$ of compatible dimensionality. By assumption it holds $\frac{\mathrm{d}\mathcal{L}}{\mathrm{d}(x_{l+1})_i} \neq 0$ for some index $i$. Then by the chain rule $x_l$ can be computed as $\left(\frac{\mathrm{d}\mathcal{L}}{\mathrm{d}(x_{l+1})_i}\right)^{-1} \cdot \left(\frac{\mathrm{d}\mathcal{L}}{\mathrm{d}(A_l)_{i,:}}\right)^T$. This allows the iterative computation of the layers' inputs as soon as the derivative of $\mathcal{L}$ w.rt. a certain layer's output is known. We conclude by noting that adding a bias can be interpreted as a layer mapping $x_k$ to $x_{k+1} = x_k + b_k$ and that $\frac{\mathrm{d}\mathcal{L}}{\mathrm{d}x_k} = \frac{\mathrm{d}\mathcal{L}}{\mathrm{d}b_k}$. □

Another interesting aspect in view of the above considerations is that many popular network architectures use fully-connected layers (or cascades thereof) as their last prediction layers. Hence the input to those prediction modules being the output of the previous layers can be reconstructed. Those activations usually already contain some information about the input image thus exposing them to attackers. For example [23] show that these features representations can be mined for image attributes by training an auxiliary malicious classifier that recognizes attributes that are not part of the main task. Further interesting in this regard is the possibility to reconstruct the ground truth label information from the gradients of the last fully-connected layer as discussed in [34]. Finally, Prop. 3.1 allows to conclude that for any classification network that ends with a fully connected layer, reconstructing the input from a parameter gradient is strictly easier than inverting visual representations, as discussed in [8, 7, 20], from their last convolutional layer.

## 4 A Numerical Reconstruction Method

As image classification networks rarely start with fully connected layers, let us turn to the numerical reconstruction of inputs: Previous reconstruction algorithms relied on two components; the euclidean cost function of Eq. (3) and optimization via L-BFGS. We argue that these choices are not optimal for more *realistic* architectures and especially *arbitrary* parameter vectors. If we decompose a parameter gradient into its norm magnitude and its direction, we find that the magnitude only captures information about the state of training, measuring local optimality of the datapoint with respect to the current model (for strongly convex functions the gradient magnitude is even an upperbound on distance to the optimal solution). In contrast, the high-dimensional direction of the gradient can carry significant information, as the angle between two data points quantifies the change in prediction at one datapoint when taking a gradient step towards another [5, 16]. As such we propose to use a cost function based on angles, i.e. cosine similarity, $l(x, y) = \langle x, y\rangle/(||x||||y||)$. In comparison to Eq. (3), the objective is not to find images with a gradient that best fits the observed gradient, but to find images that lead to a similar change in model prediction as the (unobserved!) ground truth. This is equivalent to minimizing the euclidean cost function, if one additionally constrains both gradient vectors to be normalized to a magnitude of 1.

We further constrain our search space to images within $[0, 1]$ and add only total variation [27] as a simple image prior to the overall problem, cf. [31]:

$$\arg \min_{x \in [0,1]^n} 1 - \frac{\langle \nabla_\theta \mathcal{L}_\theta(x, y), \nabla_\theta \mathcal{L}_\theta(x^*, y)\rangle}{||\nabla_\theta \mathcal{L}_\theta(x, y)||||\nabla_\theta \mathcal{L}_\theta(x^*, y)||} + \alpha \operatorname{TV}(x). \tag{4}$$

Secondly, we note that our goal of finding some inputs $x$ in a given interval by minimizing a quantity that depends (indirectly, via their gradients) on the outputs of intermediate layers, is related to the task of finding adversarial perturbations for neural networks [29, 19, 1]. As such, we minimize eq.

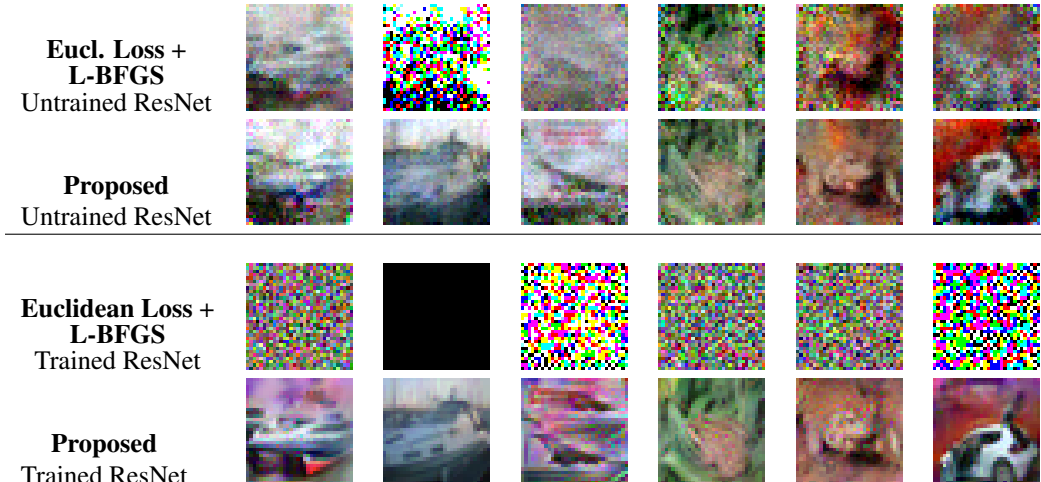

| Eucl. Loss + L-BFGS Untrained ResNet | | | | | | |
| Proposed Untrained ResNet | | | | | | |

| Euclidean Loss + L-BFGS Trained ResNet | | | | | | |
| Proposed Trained ResNet | | | | | | |

Figure 2: Baseline comparison for the network architectures shown in [31, 35].We show the first 6 images from the CIFAR-10 validation set.

(4) only based on the sign of its gradient, which we optimize with Adam [15] with step size decay. Note though that signed gradients only affect the first and second order momentum for Adam, with the actual update step still being unsigned based on accumulated momentum, so that an image can still be accurately recovered.

Applying these techniques leads to the reconstruction observed in Fig. 1. Further ablation of the proposed mechanism can be found in the appendix. We provide a `pytorch` implementation at `https://github.com/JonasGeiping/invertinggradients`.

This attack is, due to the double backpropagation, roughly twice as expensive as a single minibatch step per gradient step on the objective eq. (4). In this work, we conservatively run the attack for up to 24000 iterations, with a relatively small step size, as computational costs are not our main concern at this moment (and we assume that the attacker that is breaking privacy potentially has order-of-magnitude more computational power than the user), yet we note that smarter step size rules and larger step sizes can lead to successful attacks with a budget of only several hundred iterations.

**Remark** (Optimizing label information). *While we could also consider the label $y$ as unknown in Eq. (4) and optimize jointly for $(x, y)$ as in [35], we follow [34] who find that label information can be reconstructed analytically for classification tasks. Thus, we consider label information to be known.*

## 5    Single Image Reconstruction from a Single Gradient

Similar to previous works on breaking privacy in a federated learning setting, we first focus in the reconstruction of a single input image $x \in \mathbb{R}^n$ from the gradient $\nabla_\theta \mathcal{L}_\theta(x, y) \in \mathbb{R}^p$. This setting serves as a proof of concept as well as an upper bound on the reconstruction quality for the multi-image distributed learning settings we consider in Sec. 6. While previous works have already shown that a break of privacy is possible for single images, their experiments have been limited to rather shallow, smooth, and untrained networks. In the following, we compare our proposed approach to prior works, and conduct detailed experiments on the effect that architectural- as well as training-related choices have on the reconstruction. All hyperparameter settings and more visual results for each experiment are provided in the supp. material.

**Comparison to previous approaches.**    We first validate our approach by comparison to the Euclidean loss (3) optimized via L-BFGS considered in [31, 35, 34]. This approach can often fail due to a bad initialization, so we allow a generous setting of 16 restarts of the L-BFGS solver. For a quantitative comparison we measure the mean PSNR of the reconstruction of $32 \times 32$ CIFAR-10 images over the first 100 images from the validation set using the same shallow and smooth CNN as in [35], which we denote as "LeNet (Zhu)" as well as a ResNet architecture, both with trained and untrained parameters. Table 1 compares the reconstruction quality of euclidean loss (3) with L-BFGS optimization (as in [31, 35, 34]) with the proposed approach. The former works extremely well for

Table 1: PSNR mean and standard deviation for 100 experiments on the first images of the CIFAR-10 validation data set over two different networks with trained an untrained parameters.

| Architecture | LeNet (Zhu) | | ResNet20-4 | |
|---|---|---|---|---|
| Trained | False | True | False | True |
| Eucl. Loss + L-BFGS | **46.25 ± 12.66** | 13.24 ± 5.44 | 10.29 ± 5.38 | 6.90 ± 2.80 |
| Proposed | 18.00 ± 3.33 | **18.08 ± 4.27** | **19.83 ± 2.96** | **13.95 ± 3.38** |

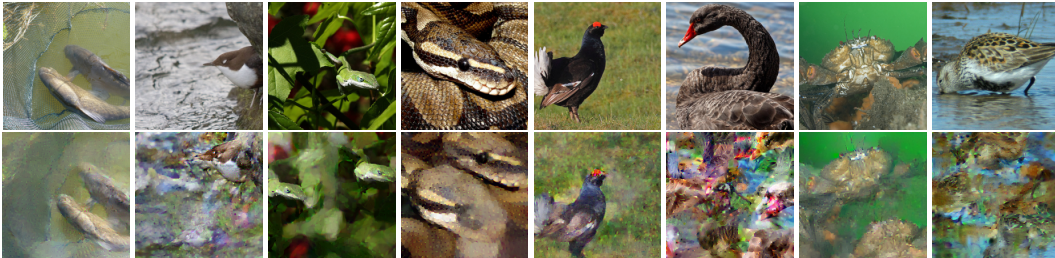

Figure 3: Single-Image Reconstruction from the parameter gradients of trained ResNet-152. Top row: Ground Truth. Bottom row: Reconstruction. We check every 1000th image of the ILSVRC2012 validation set. The amount of information leaked per image is highly dependent on image content - while some examples like the two tenches are highly compromised, the black swan (ironically) leaks almost no usable information. Noticeable is also the loss of positional information in several images.

the untrained, smooth, shallow architecture, but completely fails on the trained ResNet. We note that [31] applied a GAN to enhance image quality from the LBFGS reconstruction, which, however, fails, when the representative is too distorted to be enhanced. Our approach provides recognizable images and works particularly well on the realistic setting of a trained ResNet as we can see in Figure 2. Interestingly, the reconstructions on the trained ResNet have a better visual quality than those of the untrained ResNet, despite their lower PSNR values according to table 1. Let us study the effect of trained network parameters in an even more realistic setting, i.e., for reconstructing ImageNet images from a ResNet-152.

**Trained vs. untrained networks.** If a network is trained and has sufficient capacity for the gradient of the loss function $\mathcal{L}_\theta$ to be zero for different inputs, it is obvious that they can never be distinguished from their gradient. In practical settings, however, owing to stochastic gradient descent, data augmentation and a finite number of training epochs, the gradient of images is rarely entirely zero. While we do observe that image gradients have a much smaller magnitude in a trained network than in an untrained one, our magnitude-oblivious approach of (4) still recovers important visual information based only on the direction of the trained gradients.

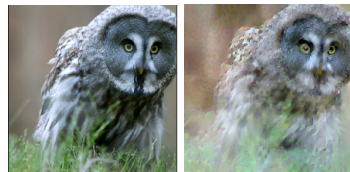

We observe two general effects on trained networks that we illustrate with our ImageNet reconstructions in Fig. 3: First, reconstructions seem to become *implicitly biased* to typical features of the same class in the training data, e.g., the more blueish feathers of the capercaillie in the 5th image, or the large eyes of the owl in the inset figure. Thus, although the overall privacy of most images is clearly breached, this effect at least obstructs the recovery of fine scale details or the image's background. Second, we find that the data augmentation used during the training of neural networks leads to trained networks that make the *localization* of objects more difficult: Notice how few of the objects in Fig. 3 retain their original position and how the snake and gecko duplicate. Thus, although image reconstruction with networks trained with data augmentation still succeeds, some location information is lost.

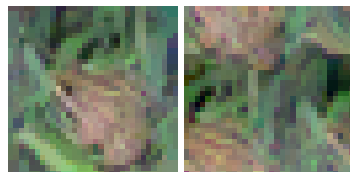

**Translational invariant convolutions.** Let us study the ability to obscure the location of objects in more detail by testing how a conventional convolutional neural network, that uses convolutions with zero-padding, compares to a provably translationally invariant CNN, that uses convolutions with circular padding. As shown in the inset figure, while the conventional CNN allows

| Original | ResNet-18 with base width: | | | ResNet-34 | ResNet-50 |
| | 16 | 64 | 128 | | |
|---|---|---|---|---|---|
| 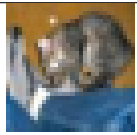 |  |  |  |  |  |
| PSNR | 17.24 | 17.37 | 25.25 | 18.62 | 21.36 |
| Avg. PSNR | 19.02 | 22.04 | 22.94 | 21.59 | 20.98 |
| Std. | 2.84 | 5.89 | 6.83 | 4.49 | 5.57 |

Figure 4: Reconstructions of the original image (left) for multiple ResNet architectures. The PSNR value refers to the displayed image while the avg. PSNR is calculated over the first 10 CIFAR-10 images. The standard deviation is the average standard deviation of one experiment under a given architecture. The ResNet-18 architecture is displayed for three different widths.

for recovery of a rather high quality image (left), the translationally invariant network makes the localization of objects impossible (right) as the original object is separated. As such we identify the common zero-padding as a source of privacy risk.

**Network Depth and Width.** For classification accuracy, the depth and number of channels of each layer of a CNN are very important parameters, which is why we study their influence on our reconstruction. Figure 4 shows that the reconstruction quality measurably increase with the number of channels. Yet, the larger network width is also accompanied with an increasing variance of experimental success. However with multiple restarts of the experiment, better reconstructions can be produced for wider networks, resulting in PSNR values that increases from 19 to almost 23 for when increasing the number of channels from 16 to 128.As such, greater network width increases the computational effort of the attacker, but does not provide greater security.

Looking at the reconstruction results we obtain from ResNets with different depths, the proposed attack degrades very little with an increased depth of the network. In particular - as illustrated in Fig. 3, even faithful ImageNet reconstructions through a ResNet-152 are possible.

## 6 Distributed Learning with Federated Averaging and Multiple Images

So far we have only considered recovery of a single image from its gradient and discussed limitations and possibilities in this setting. We now turn to strictly more difficult generalized setting of *Federated Averaging* [21, 22, 26] and *multi-image* reconstruction, to show that the proposed improvements translate to this more practical case as well, discussing possibilities and limits in this application.

Instead of only calculating the gradient of a network's parameters based on local data, federated averaging performs multiple update steps on local data before sending the updated parameters back to the server. Following the notation of [21], we let the local data on the user's side consist of $n$ images. For a number $E$ of local epochs the user performs $\frac{n}{B}$ stochastic gradient update steps per epoch, where $B$ denotes the local mini-batch size, resulting in a total number of $E\frac{n}{B}$ local update steps. Each user $i$ then sends the locally updated parameters $\tilde{\theta}_i^{k+1}$ back to the server, which in turn updates the global parameters $\theta^{k+1}$ by averaging over all users.

We empirically show that even the setting of federated averaging with $n \geq 1$ images is potentially amenable for attacks. To do so we try to reconstruct the local batch of $n$ images by the knowledge of the local update $\tilde{\theta}_i^{k+1} - \theta^k$. In the following we evaluate the quality of the reconstructed images for different choices of $n$, $E$ and $B$. We note that the setting studied in the previous sections corresponds to $n = 1$, $E = 1$, $B = 1$. For all our experiments we use an untrained ConvNet.

**Multiple gradient descent steps, $B = n = 1$, $E > 1$:**
Fig. 5 shows the reconstruction of $n = 1$ image for a varying number of local epochs $E$ and different choices of learning rate $\tau$. Even for a high number of 100 local gradient descent steps the reconstruction quality is unimpeded. The only failure case we were able to exemplify was induced by picking a high learning rate of 1e-1. This setup, however, corresponds to a step size that would lead to a divergent training update, and as such does not provide useful model updates.

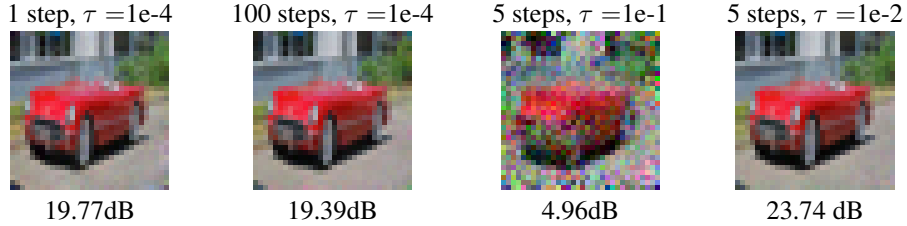

| 1 step, $\tau =$1e-4 | 100 steps, $\tau =$1e-4 | 5 steps, $\tau =$1e-1 | 5 steps, $\tau =$1e-2 |
|:---:|:---:|:---:|:---:|
| 19.77dB | 19.39dB | 4.96dB | 23.74 dB |

Figure 5: Illustrating the influence of the number of local update steps and the learning rate on the reconstruction: The left two images compare the influence of the number of gradient descent steps for a fixed learning rate of $\tau =$1e-4. The two images on the right result from varying the learning rate for a fixed number of 5 gradient descent steps. PSNR values are shown below the images.

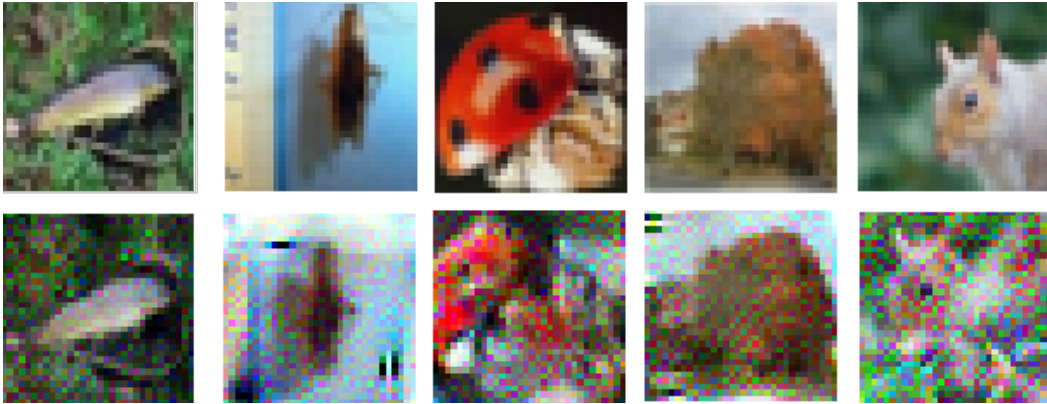

Figure 6: Information leakage from the aggregated gradient of a batch of 100 images on CIFAR-100 for a ResNet32-10. Shown are the 5 *most* recognizable images from the whole batch. Although most images are unrecognizable, privacy is broken even in a large-batch setting. We refer to the supplementary material for all images.

**Multi-Image Recovery,** $B = n > 1$, $E = 1$:
So far we have considered the recovery of a single image only, and it seems reasonable to believe that averaging the gradients of multiple (local) images before sending an update to the server, restores the privacy of federated learning. While such a multi-image recovery has been considered in [35] for $B \leq 8$, we demonstrate that the proposed approach is capable of restoring some information from a batch of 100 averaged gradients: While most recovered images are unrecognizable (as shown in the supplementary material), Fig. 6 shows the 5 most recognizable images and illustrates that even averaging the gradient of 100 images does not entirely secure the private data. Most surprising is that the distortions arising from batching are *non-uniform*. One could have expected all images to be equally distorted and near-irrecoverable, yet some images are highly distorted and others only to an extend at which the pictured object can still be recognized easily, which demonstrates that privacy leaks are conceivable even for large batches of image data.

Note that the attacker in this scenario only has knowledge about the average of gradients, however we assume the number of participating images to be known to the server. The server might request this information anyway (for example to balance heterogeneous data), but even if the exact number of images is unknown, the server (which we assume to have significantly more compute power than the user) could run reconstructions over a range of candidate numbers, given that the number of images is only a small integer value and then select the solution with minimal reconstruction loss.

**General case**
We also consider the general case of multiple local update steps using a subset of the whole local data in each mini batch gradient step. An overview of all conducted experiments is provided in Table 2. For each setting we perform 100 experiments on the CIFAR-10 validation set. For multiple images in a mini batch we only use images of different labels avoiding permutation ambiguities of reconstructed images of the same label. As to be expected, the single image reconstruction turns out to be most

Table 2: PSNR statistics for various federated averaging settings, averaged over experiments on the first 100 images of the CIFAR-10 validation data set.

| 1 epoch | | | 5 epochs | |
|---|---|---|---|---|
| 4 images | 8 images | | 1 image | 8 images |
| batchsize 2 | batchsize 2 | batchsize 8 | batchsize 1 | batchsize 8 |
| $16.92 \pm 2.10$ | $14.66 \pm 1.12$ | $16.49 \pm 1.02$ | $25.05 \pm 3.28$ | $16.58 \pm 0.96$ |

amenable to attacks in terms of PSNRs values. Despite a lower performance in terms of PSNR, we still observe privacy leakage for all multi-image reconstruction tasks, including those in which gradients in random mini-batches are taken. Comparing the full-batch, 8 images examples for 1 and 5 epochs, we see that our previous observation that multiple epochs do not make the reconstruction problem more difficult, extends to multiple images. For a qualitative assessment of reconstructed images of all experimental settings of Table 2, we refer to the supplementary material.

# 7 Conclusions

Federated learning is a modern paradigm shift in distributed computing, yet its benefits to privacy are not as well understood yet. We shed light into possible avenues of attack, analyze the ability to reconstruct the input to any fully connected layer analytically, propose a general optimization-based attack based on cosine similarity of gradients, and discuss its effectiveness for different types of architectures and scenarios. In contrast to previous work we show that even *deep*, *nonsmooth* networks trained with ImageNet-sized data such as modern computer vision architectures like ResNet-152 are vulnerable to attacks - even when considering *trained* parameter vectors. Our experimental results clearly indicate that privacy is not an innate property of collaborative learning algorithms like federated learning, and that secure applications to be closely investigated on a case-by case basis for their potential of leaking private information. Provable differential privacy possibly remains the only way to *guarantee* security, even for aggregated gradients of larger batches of data points.

## Broader Impact - Federated Learning does not guarantee privacy

Recent works on privacy attacks in federated learning setups ([25, 24, 31, 35, 34]) have hinted at the fact that previous hopes that "Privacy is enhanced by the ephemeral and focused nature of the [Federated Learning] updates" [3] are not true in general. In this work, we demonstrated that improved optimization strategies such as a cosine similarity loss and a signed Adam optimizer allow for image recovery in a federated learning setup in industrially realistic settings for computer vision: Opposed to the idealized architectures of previous works we demonstrate that image recovery is possible in deep, non-smooth and trained architectures over multiple federated averaging steps of the optimizer and even in batches of 100 images.

We note that image classification is possibly especially vulnerable to these types of attacks, given the inherent structure of image data, the size of image classification networks, and the comparatively small number of images a single user might own, relative to other personal information. On the other hand, this attack is likely only a first step towards stronger attacks. Therefore, this work points out that the question how to protect the privacy of our data while collaboratively training highly accurate machine learning approaches remains largely unsolved: While differential privacy offers provable guarantees, it also reduces the accuracy of the resulting models significantly [12]. As such differential privacy and secure aggregation can be costly to implement so that there is some economic incentive for data companies to use only basic federated learning. For a more general discussion, see [30]. There is strong interest in further research on privacy preserving learning techniques that render the attacks proposed in this paper ineffective. This might happen via defensive mechanisms or via computable guarantees that allow practitioners to verify whether their specific application is vulnerable to such an attack and within which bounds.

## Acknowledgments and Disclosure of Funding

This research was directly supported by the University of Siegen. HB and MM further received support from the German Research Foundation (DFG) under grant MO 2962/2-1.

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
