[Supplementary Material · supp_inverting_gradients.pdf]

# Supplementary Material:
# Inverting Gradients - How easy is it to break privacy in federated learning?

## Abstract

This supp. material discusses variations of the threat model in Sec. A. It details the variant used to attack federated averaging (Sec. 6 in the main paper) and ConvNet architecturein Sec. B. Further, hyperparameter settings for all experiments and visual representations of the results of Section 4.2 are recorded in C. A proof of proposition 3.1 of the main paper is included in Sec.D, and finally more examples for ImageNet-scale images and the full 100 images for the multi-image experiment of Sec. 6.1 are shown in Sec. E.

## A  Variations of the threat model

In this work we consider a *honest-but-curious* threat model as discussed in the introduction. Straying from this scenario could be done primarily in two ways: First by changing the architecture, and second by keeping the architecture non-malicious, but changing the global parameters sent to the user.

### A.1  Dishonest Architectures

So far we assumed that the server operates under an *honest-but-curious* model, and as such would not modify the model maliciously to make reconstruction easier. If we instead allow for this, then reconstruction becomes nearly trivial: Several mechanisms could be used: Following Prop. 1, the server could, for example, place a fully-connected layer in the first layer, or even directly connect the input to the end of the network by concatenation. Slightly less obvious, the model could be modified to contain reversible blocks [1, 3]. These blocks allow the recovery of input from their outputs. From Prop. 1 we know that we can reconstruct the input to the classification layer, so this allows for immediate access to the input image. If the server maliciously introduces separate weights or sub-models for each batch example, then this also allows for a recovery of an arbitrarily large batch of data. Operating in a setting, where such behavior is possible would require the user (or a provider trusted by the user) to vet any incoming model either manually or programmatically.

### A.2  Dishonest Parameter Vectors

However, even with a fixed *honest* architecture, a malicious choice of global parameters can significantly influence reconstruction quality. For example, considering the network architecture in [6] which does not contain strides and flattens convolutional features, the dishonest server could set all convolution layers to represent the identity [2], moving the input through the network unchanged up to the classification layer, from which the input can be analytically computed as in Prop. 1. Likewise for

| 5.9e-1 | 29.37dB | 4.6e+2 | 26.62dB | 1.8e+2 | 27.37dB | 1.5e+2 | 18.27dB |
|---|---|---|---|---|---|---|---|

Figure 1: Label flipping. Images can be easily reconstructed when two rows in the parameters of the final classification layer are permuted. Below each input image is given the gradient magnitude, below each output image its PSNR. Compare these results to the additional examples in Fig. 3

an architecture that contains strides to a recognizable lower resolution [5], the input can be recovered immediately albeit in a smaller resolution when the right parameter vector is sent to the user.

Such a specific choice of parameters is however likely detectable. A subtler approach, as least possible in theory, would be to optimize the network parameters themselves that are sent to the user so that reconstruction quality from these parameters is maximized. While such an attack is likely to be difficult to detect on the user-side, it would also be very computationally intensive.

**Label flipping.** There is even a cheaper alternative. According to Sec. 5, very small gradient vectors may contain less information. A simple way for a dishonest server to boost these gradients is to permute two rows in the weight matrix and bias of the classification layer, effectively flipping the semantic meaning of a label. This attack is difficult to detect for the user (as long as the gradient magnitude stays within usual bounds), but effectively tricks him into differentiating his network w.r.t to the wrong label. Fig. 1 shows that this mechanism can allow for a reliable reconstruction with boosted PSNR scores, as the effect of the trained model is negated.

# B  Experimental Details

| 3 x 3 Conv, 1 * D |
| 3 x 3 Conv, 2 * D |
| 3 x 3 Conv, 2 * D |
| 3 x 3 Conv, 4 * D |
| 3 x 3 Conv, 4 * D |
| 3 x 3 Conv, 4 * D |
| MaxPool2D(3) |
| 3 x 3 Conv, 4 * D |
| 3 x 3 Conv, 4 * D |
| 3 x 3 Conv, 4 * D |
| MaxPool2D(3) |
| FC, 10 |

Figure 2: Network architecture *ConvNet*, consisting of 8 convolution layers, specified with corresponding number of output channels. Each convolution layer is followed by a batch normalization layer and a ReLU layer. $D$ scales the number of output channels and is set to $D = 64$ by default.

## B.1  Federated Averaging

The extension of Eq. (4) to the case of federated averaging (in which multiple local update steps are taken and sent back to the server) is straightforward. Notice first, that given old parameters $\theta^k$, local updates $\theta^{k+l}$, learning rate $\tau$, and knowledge about the number of update steps[1], the update can be

Table 1: Ablation Study for the proposed approach for a trained ResNet-18 architecture, trained on CIFAR-10. Reconstruction PSNR scores are averaged over the first 10 images of the CIFAR-10 validation set (Standard Error in parentheses).

| Basic Setup | 20.12 dB ($\pm 1.02$) |
|---|---|
| L2 Loss instead of cosine similarity | 15.13 dB ($\pm 0.70$) |
| Without total variation | 19.96 dB ($\pm 0.75$) |
| With L-BFGS instead of Adam | 5.13 dB ($\pm 0.50$) |

49  rewritten as the average of updated gradients.

$$\theta^{k+l} = \theta^k - \tau \sum_{m=1}^{l} \nabla_{\theta^{k+m}} \mathcal{L}_{\theta^{k+m}}(x, y) \tag{1}$$

50  Subtracting $\theta^k$ from $\theta^{k+l}$, we simply apply the proposed approach to the resulting average of updates:

$$\arg \min_{x \in [0,1]^n} 1 - \frac{\langle \sum_{m=1}^{l} \nabla_{\theta^{k+m}} \mathcal{L}_{\theta^{k+m}}(x, y), \sum_{m=1}^{l} \nabla_{\theta^{k+m}} \mathcal{L}_{\theta^{k+m}}(x^*, y) \rangle}{|| \sum_{m=1}^{l} \nabla_{\theta^{k+m}} \mathcal{L}_{\theta^{k+m}}(x, y)|| || \sum_{m=1}^{l} \nabla_{\theta^{k+m}} \mathcal{L}_{\theta^{k+m}}(x^*, y)||} + \alpha \, \mathrm{TV}(x). \tag{2}$$

51  Using automatic differentiation, we backpropagate the gradient w.r.t to $x$ from the average of update
52  steps.

### B.2  ConvNet

54  We use a ConvNet architecture as a baseline for our experiments as it is relatively fast to optimize,
55  reaches above 90% accuracy on CIFAR-10 and includes two max-pooling layers. It is a rough
56  analogue to AlexNet [4]. The architecture is described in Fig. 2.

### B.3  Ablation Study

58  We provide an ablation for proposed choices in Table 1. We note that two things are central, the
59  Adam optimizer and the similarity loss. Total variation is a small benefit, and using signed gradients
60  is a minor benefit.

## C  Hyperparameter Settings

62  In our experiments we reconstruct the network's input using Adam based on signed gradients as
63  optimization algorithm and cosine similarity as cost function as described in Sec. 4. It is important
64  to note that the optimal hyperparameters for the attack depend on the specific attack scenario -
65  that the attack fails with default parameters is no guarantee for security. We always initialize our
66  reconstructions from a Gaussian distribution with mean 0 and variance 1 (Note that the input data is
67  normalized as usual for all considered datasets) and set the step size of the optimization algorithm
68  within $[0.01, 1]$. We use a smaller step sizes of $0.1$, for the wider and deeper networks in Sec. 5.2
69  and a larger step sizes of $1$ for the federated averaging experiments in Sec 6, with $0.1$ being the
70  default choice. The optimization runs for up to 24000 iterations. The step size decay is always fixed,
71  occuring after $\frac{3}{8}$, $\frac{5}{8}$ and $\frac{7}{8}$ of iterations and reducing the learning rate by a factor of $0.1$ each time. The
72  number of iterations is a generally conservative estimate, privacy can often be broken much earlier.

73  We tweak the total variation parameter depending on the specific attack scenario, however note that
74  its effect on avg. PSNR is mostly minor as seen in table 1. When not otherwise noted we default to a
75  value of $0.01$.

76  **Remark** (Restarts). *Generally, multiple restarts of the attack from different random initializations can*
77  *improve the attack success moderately. However they also increase the computational requirements*
78  *significantly. To allow for quantitative experimental evaluations of multiple images, we do not*
79  *consider restarts in this work (aside from Sec. 5 where we apply them to improve results of the*
80  *competing LBFGS solver) - but stress that an attacker with enough ressources could further improve*
81  *his attack by running it with multiple restarts.*

| Architecture | LeNet (Zhu) | | ResNet20-4 | |
|---|---|---|---|---|
| Trained | False | True | False | True |
| TV | $10^{-2}$ | $10^{-3}$ | 0 | $10^{-2}$ |

Table 2: TV regularization values used for the proposed approach in the baseline experiments of Section 5.

| Number of epochs $E$ | 1 | 1 | 1 | 5 | 5 |
|---|---|---|---|---|---|
| Number of local images $n$ | 4 | 8 | 8 | 1 | 8 |
| Mini-batch size $B$ | 2 | 2 | 8 | 1 | 8 |
| TV | $10^{-6}$ | $10^{-6}$ | $10^{-4}$ | $10^{-4}$ | $10^{-4}$ |

Table 3: Total variation weights for the reconstruction of network input in the experiments in Sec. 4.2

## C.1 Settings for the experiments in Sec. 5

**Comparison to previous approaches** For comparison with baselines in section 5, we re-implement the network from [6], which we dub LeNet (Zhu) in the following, and additionally run all experiments for the ResNet20-4 architecture. We base both the network and the approach on code from the authors of [6], [2]. For the LBFGS-L2 optimization we use a learning rate of $1e-4$ and 300 iterations. For the ResNet experiments we use the generous amount of 8 restarts and for the faster to optimize LeNet (Zhu) architecture we use the even higher number of 16 restarts. All experiment conducted with the proposed approach only use one restart, 4800 iterations, a learning rate of 0.1 and TV regularization parameters as detailed in Table 2. Note that in the described settings the proposed method took significantly less time to optimize than the LBFGS optimization.

**Spatial Information** The experiments on spatial information are performed on the ConvNet architecture with $D = 64$ channels.

## C.2 Setting for experiments in Sec. 6

For the five cases consider in Table 2 we consider an untrained ConvNet, a learning rate of 1, 4800 iterations, one restart and the TV regularization parameters as given in table 3. Each of the 100 experiments uses different images, i.e. each experiments uses the images of the CIFAR-10 validation set following the ones used in the previous experiment. As multiple images of the same label in one mini-batch cause an ambiguity in the ordering of images w.r.t. that label, we do not consider that case. If an image with an already encountered label is about to be added to the respective mini-batch we skip that image and use the next image of the validation set with a different label.

# D Proofs for section 3.1

In the following we give a more detailed proof of Prop 3.1, which is follows directly from the two propositions below:

**Proposition D.1.** *Let a neural network contain a biased fully-connected layer at some point, i.e. for the layer's input $x_l \in \mathbb{R}^{n_l}$ its output $x_{l+1} \in \mathbb{R}^{n_{l+1}}$ is calculated as $x_{l+1} = \max\{y_l, 0\}$ for*

$$y_l = A_l x_l + b_l, \tag{3}$$

*for $A_l \in \mathbb{R}^{n_{l+1} \times n_l}$ and $b_l \in \mathbb{R}^n_{l+1}$. Then the input $x_l$ can be reconstructed from $\frac{\mathrm{d}\mathcal{L}}{\mathrm{d}A_l}$ and $\frac{\mathrm{d}\mathcal{L}}{\mathrm{d}b_l}$, if there exists an index $i$ s.t. $\frac{\mathrm{d}\mathcal{L}}{\mathrm{d}(b_l)_i} \neq 0$.*

*Proof.* It holds that $\frac{d\mathcal{L}}{d(b_l)_i} = \frac{d\mathcal{L}}{d(y_l)_i}$ and $\frac{dy_i}{d(A_l)_{i,:}} = x^T$. Therefore

$$\frac{d\mathcal{L}}{d(A_l)_{i,:}} = \frac{d\mathcal{L}}{d(y_l)_i} \cdot \frac{d(y_l)_i}{d(A_l)_{i,:}} \tag{4}$$

$$= \frac{d\mathcal{L}}{d(b_l)_i} \cdot x_l^T \tag{5}$$

for $(A_l)_{i,:}$ denoting the $i^{\text{th}}$ row of $A_l$. Hence $x_l$ can can be uniquely determined as soon as $\frac{d\mathcal{L}}{d(b_l)_i} \neq 0$. □

**Proposition D.2.** *Consider a fully-connected layer (not necessarily including a bias) followed by a ReLU activation function, i.e. for an input $x_l \in \mathbb{R}^{n_l}$ the output $x_{l+1} \in \mathbb{R}^{n_{l+1}}$ is calculated as $x_{l+1} = \max\{y_l, 0\}$ for*

$$y_l = A_l x_l, \tag{6}$$

*where the maximum is computed element-wise. Now assume we have the additional knowledge of the derivative w.r.t. to the output $\frac{d\mathcal{L}}{dx_{l+1}}$. Furthermore assume there exists an index $i$ s.t. $\frac{d\mathcal{L}}{d(x_{l+1})_i} \neq 0$. Then the input $v$ can be derived from the knowledge of $\frac{d\mathcal{L}}{dA_l}$.*

*Proof.* As $\frac{d\mathcal{L}}{d(x_{l+1})_i} \neq 0$ it holds that $\frac{d\mathcal{L}}{d(y_l)_i} = \frac{d\mathcal{L}}{d(x_{l+1})_i}$ and it follows that

$$\frac{d\mathcal{L}}{d(A_l)_{i,:}} = \frac{d\mathcal{L}}{d(y_l)_i} \cdot \frac{d(y_l)_i}{d(A_l)_{i,:}} \tag{7}$$

$$= \frac{d\mathcal{L}}{d(x_{l+1})_i} \cdot x_l^T. \tag{8}$$

□

# E  Additional Examples

## E.1  Additional CIFAR-10 examples

Figure 3 shows additional "extreme" examples for CIFAR-10, reconstructing the image with lowest and the image with largest gradient magnitude for the training and validation set of CIFAR-10 for trained and untrained ConvNet and ResNet20-4 models.

## E.2  Visualization of experiments in Sec. 5

**Network Width**  The reconstructions for the first six CIFAR images for different width ResNet-18 architectures are given in Fig. 4.

**Network Depth**  The experiments concerning the network depth are performed for different deep ResNet architectures. Multiple reconstruction results for different deep networks are shown in Fig. 5.

## E.3  More ImageNet examples for Sec. 5

Fig. 6 shows further instructive examples of reconstructions for ImageNet validation images for a trained ResNet-18 (the same setup as Fig. 3 in the main paper). We show a very good reconstruction (German shepherd), a good, but translated reconstruction (giant panda) and two failure cases (ambulance and flower). For the ambulance, for example, the actual writing on the ambulance car is still hidden. For the flower, the exact number of petals is hidden. Also, note how the reconstruction of the giant panda is much clearer than that of the tree stump co-occurring in the image, which we consider an indicator of the self-regularizing effect described in Sec. 5.

Figures 7 and 8 show more examples. We note that the examples in these figures and in Figure 3 are not handpicked, but chosen neutrally according to their ID in the ILSVRC2012, ImageNet, validation set. The ID for each image is obtained by sorting the synset that make up the dataset in increasing order according to their synset ID and sorting the images within each synset according to their synset ID in increasing order. This is the default order in `torchvision`.

**Trained ConvNet**

Images from the training set · Images from the validation set

| 4.5e-21 | 18.04dB | 2.5e+02 | 14.85dB | 9.8e-17 | 14.60dB | 5.5e+02 | 30.26dB |

**Trained ResNet20-4**

Images from the training set · Images from the validation set

| 5.3-06 | 15.21dB | 1.0e+2 | 19.75dB | 1.2e-5 | 13.84dB | 4.6e+2 | 15.53dB |

**Untrained ConvNet** · **Untrained ResNet20-4**

| 6.1e-1 | 31.36dB | 6.7e-1 | 31.16dB | 3.8e+1 | 21.90dB | 4.5e+1 | 20.23dB |

Figure 3: Reconstruction of images for the *trained* ConvNet model (Top) and ResNet20-4 (middle). We show reconstructions of the **worst-case** image and **best case** image from CIFAR-10, based on gradient magnitude for both the training and the validation set. Below each input image is given the gradient magnitude, below each output image its PSNR. The bottom row shows reconstructions for the worst-case examples for untrained models.

16 Channels

64 Channels

128 Channels

Figure 4: Reconstructions using ResNet-18 architectures with different widths.

ResNet-18

ResNet-34

ResNet-50

Figure 5: Reconstructions using different deep ResNet architectures.

Figure 6: Additional qualitative ImageNet examples, failure cases and positive cases for a trained ResNet-18. Images taken from the ILSVRC2012 validation set.

Figure 7: Additional single-image reconstruction from the parameter gradients of trained ResNet-152. Top row: Ground Truth. Bottom row: Reconstruction. The paper showed images 0000, 1000, 2000, 3000, 4000, 5000, 6000, 7000 from the ILSVRC2012 validation set. These are images 8000-12000.

Figure 8: Additional single-image reconstruction from the parameter gradients of trained ResNet-152. Top row: Ground Truth. Bottom row: Reconstruction. These are images 500, 1500, 2500, 3500, 4500.

Figure 9: Results of the first 100 experiments for $E = 5$, $n = 1$, $B = 1$.

## E.4 Multi-Image Recovery of Sec. 6

For multi-image recovery, we show the full set of 100 images in Fig. 14, we recommend to zoom in to a digital version of the figure. The success rate for separate images is semi-random, depending on the initialization.

## E.5 General case of Sec. 6

We show the results for the first ten experiments in Figures 9, 10, 11, 12, 13. In Figure 9 we even show all 100 experiments as there only one image is used per experiment.

Additional images are following on the next pages.

Figure 10: Results of the first ten experiments for $E = 1$, $n = 4$, $B = 2$.

Figure 11: Results of the first ten experiments for $E = 1$, $n = 8$, $B = 2$.

Figure 12: Results of the first ten experiments for $E = 1$, $n = 8$, $B = 8$.

Figure 13: Results of the first ten experiments for $E = 5$, $n = 8$, $B = 8$.

Figure 14: Full results for the batch of CIFAR-100 images. Same experiment as in Fig. 6 of the paper.

## Footnotes

[1]We assume that the number of local updates is known to the server, yet this could also be found by brute-force, given that $l$ is a small integer.

[2] https://github.com/mit-han-lab/dlg