[Reviews · NeurIPS 2020]

Review 1

Summary and Contributions: This paper studies example image reconstruction from loss gradients for deep networks. It gives experimental evidence that reconstruction is possible and uses this to suggest that federated learning does not necessarily provide meaningful privacy guarantees. The main claimed advance is that previous results to this effect were only demonstrated for architectures with a smaller number of layers.

Strengths: Caveat: I am familiar with the privacy literature, especially differential privacy, but I am almost entirely unfamiliar with deep learning. As such, my review is low confidence. Strengths: As an outsider to the area, federated learning's privacy claims have appeared a little suspicious to me, so I appreciate a work that aims to test these claims. Taking the author's claims of reasonable experimental setup at face value (see caveat above), the results look pretty impressive to me. As shown in the Supplement, image recovery is still inconsistent at best, but less-than-consistent recovery is in my opinion still good enough evidence that in practice loss gradients really are not intrinsically private objects.

Weaknesses: Weaknesses: While the evidence that the method “works” seems reasonable, t’s hard to tell whether this paper represents a significant advance over existing work on image reconstruction from gradients. For example, in the related work, the paper states that “[35] also discuss deeper architectures, but provide no tangible results”. Is this an intrinsic failing of their method that this paper improves, or did they simply not conduct those experiments? The only empirical comparison appears to be the PSNR comparison in Table 1, but then Line 185 observes that PSNR doesn’t correlate perfectly with whether the image reconstruction looks convincing. In that sense, the reconstruction images themselves seem like the most useful point of comparison, so it’s disappointing that the reconstruction images aren’t compared directly. The formal result, Proposition 3.1, seems out of place — are neural networks consisting solely of fully connected layers common? The preamble to this result also says that it’s a generalization of a textbook example, which seems odd as well.

Correctness: Given my unfamiliarity with this area, I don't think I can really assess the empirical methodology of this paper.

Clarity: I don't see any major clarity issues.

Relation to Prior Work: As mentioned in "Weaknesses", I'd like a better empirical comparison to past methods. Right now this is limited to Table 1 in the main body and includes no actual image reconstructions. How does past work (e.g. [35] in the paper) do on the examples presented here?

Reproducibility: Yes

Additional Feedback: My main comments and questions appear in "Weaknesses" above. --- Update after author response: I think the paper is a solid contribution, but my unfamiliarity with the area means I'll keep my score at 6, with a bit more confidence. In my opinion, the main contribution of this paper is its reconstruction results and not so much its technical novelty. For that reason, I think the paper would be stronger if it expanded Section 5 and reduced Section 3. In particular, more material like that of Figure 2 would make a better case for the superior results obtained through the paper's new approach. If the author's really can't find the space, at least include it in the Supplement.


Review 2

Summary and Contributions: This paper designs a new method to recover the training data from the gradient in federated learning systems. It first theoretically proves that the input to a fully connected layer can be reconstructed independent of the architecture. Then it adopts a different loss function and optimization solution to recover images. Comprehensive evaluations indicate that their solution has better recovery results than prior works.

Strengths: + Comprehensive evaluations considering different perspectives, e.g., single image, batch, federated averaging, etc. + Better attack results than past solutions.

Weaknesses: - The technical novelty is minor - The attacks results for batch training seem not good as claimed. - The theoretical analysis for fully-connected layer is not quite meaningful.

Correctness: Yes

Clarity: Yes

Relation to Prior Work: Yes

Reproducibility: Yes

Additional Feedback: This paper gives a very clear description about the status quo of model inversion attacks in federated learning. Also it gives a very comprehensive evaluation about the new privacy attack solution considering different settings and configurations. However, this paper also has some limitations. 1. The technical novelty is very minor. Compared to prior works, the only modification the authors proposed is changing the cost function from Euclidean distance to cosine similarity, and changing the optimization solver from L-BFGS to Adam. These are very common methods. The entire attack idea, scenario and techniques are the same. Such contribution seems too small for NeurIPS venue. 2. The theoretical analysis about attacking fully-connected layers is interesting. However, it seems very useless. In reality, it is very rare to just use fully-connected layers for classification. Recovering the input of FC layers does not leak very critical information. The authors claimed in Lines 125 - 133 that such input is meaningful. However, they do not give a very realistic scenario, or empirical evidence. Thus, this cannot convince me. 3. Recovery of the batch results does not seem very good visually. The authors claimed that the proposed technique can recover batch images with the size of 100. This is much more significant than prior work (8 in [35]). But checking Figure 6, even the 5 most recognizable images are hard to recognize, and the quality of recovered images is not satisfactory. In comparison, the attack results in Figure 4 of [35] are much better. So I have doubt about the claim that the attack is effective for batch size of 100. What is the criteria for a good recovered images? Other settings in this paper all report quantitative results (PSNR), but this one does not have (either in the main paper or supplementary material). Post-rebuttal: Although this paper has limited novelty, the results are indeed good and promising.


Review 3

Summary and Contributions: This paper proposed a new optimization based reconstruction attack against machine learning models (and federated learning) with faithful results. It’s shown that by collecting the gradients information, a potential attacker can successfully reconstruct images even in a much more practical setting, i.e., deeper and wider networks on ImageNet, the results are promising and impressive, raising the issue that solely federated learning itself cannot well protect the privacy.

Strengths: 1. The proposed loss in the Numerical Reconstruction section is simple and straightforward to understand, I like the connection the authors raised between this attack and typical adversarial attack, which may bridge these two areas, leading stronger adversarial attacks to privacy attacks area. 2. The results and reconstructed images in ImageNet are faithful and impressive to me, it indeed shows that there's not much privacy in federated learning itself by exploiting the gradients information. Th attack seems much stronger and make the results easy to understand and can see directly, in contrast to membership inference attacks and attribute inference attacks, which sometimes cannot give good attack results. 3. The attack setting is clear and easy, I believe this attack methodology will help practitioners to better evaluate the privacy of trained models in industry, and may be combined with differential privacy to better understand the epsilon bounds.

Weaknesses: 1. There are some theoretical results about fully connected layers developed based on previous work, however, the authors doesn't seem to extend this result to convolution network, nor conducted experiments on fully connected layers, like experiments on reconstruct the inputs to a fully connect layer, what's the computation cost? 2. In line 138-142, the authors talk about the different effects of magnitude and direction of the gradients, i.e., magnitude only captures information about the state of training, the high-dimensional direction of the gradient can carry significant information, is this your feelings or do you have any justifications for this statement? It will be better if there are some experiments results on exploring this statement. 3. The topic of this paper I learnt from the title is on attack of federated learning model, however, the majority of the experiments and descriptions on the vanilla machine learning models (classification task), there are lack of an extensive experiments on federated learning settings, for example, including not just one participant, but many participants, and try to reconstruct each participants' data using this attack. 4. the attack setting on averaged gradients from multiple images is not clear, as I understand, for one image setting, the attack just initialize with one random point, and do the optimization using its gradients, for multiply images, the attack will initialize multiply random points and average the gradients then optimize the loss? is my understand correct? Here the attacker can only access the averaged gradients, how do you know how many images participated in this gradient? This part is important and much more realistic, but it's not clear as desired, nor can I find it in the supplementary. 5. As mentioned in several places in the paper, a deeper network doesn't make the model much more private, but only make the computation cost of the attack larger, this observation is interesting and worthy exploring, can you include a section about the specific computation cost somewhere, although the attack seems strong, the authors still need to make it clear, among A images, they get B images reconstructed with high quality, and what's the corresponding computation cost.

Correctness: The method looks correct to me, some claims are lack of justifications, as mention above, the empirical methodology also works as desired after running the code through.

Clarity: Generally the paper is well written, but with some problems: 1. The citation link doesn’t work, clicking can't take reader to the reference place. 2. Line 98, if-> whether 3. Iine 100, two ‘is’ here, remove one 4. table 1 caption, trained and untrained 5. some images may need to be reorganized, with index and caption, such as the image in line 206 place.

Relation to Prior Work: Differences are clear, the contributions are different and evident from previous work.

Reproducibility: Yes

Additional Feedback: I have read the feedback and my opinion hasn't changed. --------------------------------- 1. does the author have any idea why the reconstruction attack on CIFAR untrained model is not good as previous Eucl. Loss + L-BFGS method? 2. Since this attack works well for realistic models, would be better if the author can include attack results on an extensive degree of federated learning settings, on CIFAR and ImageNet with more participants. 3. As mentioned above, the author can try to reconstruct a model which is trained using differential privacy, to help understand the privacy epsilon, curious what will the results be. 4. there is a related paper [1] from defense perspective, using separated DP, which provide privacy on the magnitude and the direction separately, provide protection against reconstruction attacks, it would be better if some results against this method can be presented to make the results more convincing. [1] Protection Against Reconstruction and Its Applications in Private Federated Learning


Review 4

Summary and Contributions: The paper addresses the problem of privacy in collaborative learning. It designs an attack that can reconstruct input images from gradient updates where previous methods failed. This is an important finding that is supported by thorough evaluation for different settings.

Strengths: The problem addressed in this paper is very important and increasingly relevant. The results are impressive and quite surprising, such as the figure 6. showing reconstructions from relatively large batch of images. The approach is well defined, methodological and explores also realistic scenarios. The paper contains theoretical analysis as well as detailed experimental section. The paper is well written and interesting to read. The contributions are clearly stated and the importance is discussed and clear.

Weaknesses: Could not find any but at the same time this submission is not exactly my area of expertise.

Correctness: The theoretical analysis appears to be correct. It is great that realistic scenarios are studied. The empirical evaluation is very comprehensive.

Clarity: The paper is very clear.

Relation to Prior Work: The submission is discussed in the context of prior art.

Reproducibility: Yes

Additional Feedback: ------- Post rebuttal ------- After reading other reviews and rebuttal, I agree that the level of technical contribution over previous works is rather simple. However, I still find experimental results very impressive.

[Author Response · NeurIPS 2020]

**Author Response for: "Inverting Gradients - How easy is it to break privacy in federated learning"**

**General Comments:** We thank all reviewers for their valuable feedback and interest in this attack. We want to stress
that the key points of this work are a surprisingly effective new attack, evaluation of previous work in realistic settings
and attack of multi-step federated learning. To the best of our knowledge, our work is the first to investigate the
multi-step setting at all.
Some questions arose about the theoretical analysis for fully connected layers. This analysis is not only meaningful in
other domains, such as medicine (e.g. Jarrar et al., "MLP neural network classifiers for medical image segmentation")
and financial manners (e.g. Kadhim et al. "Prediction of the Performance Related to Financial Capabilities"), but also
shows that attacks on gradient data are easier than inversion from feature representations as in [20]. Also, this result
directly applies to iRevNets (which have reversible feature representations), showing that even deep CNN architectures
exist that leak all information. Finally knowledge of the feature representation already enables attacks like Melis et al.
[23], which utilize an auxiliary malicious classifier.
**Reviewer 1:** Note that our main claim is not only that previous results did only consider shallower networks, but
also that previous results consider unrealistic settings (smooth activations, no stride in convolutional layers, untrained
parameters), which make reconstruction easier. Our results show that a stronger attack still succeeds in a realistic setting
both for a single image, and for multiple images and steps. We directly compare visually to previous works (which both
use L-BFGS and Euclidean loss) in Fig.2, showing that previous works struggle in realistic settings. We will extend our
supplementary material and show additional visual results for Table 1.
**Reviewer 2:** Regarding technical novelty, note that we propose a deceptively simple new attack that nevertheless
significantly broadens the applicability of gradient inversion attacks. We formalize a realistic threat scenario and
evaluate previous works and the new attack in this new setting. We investigate why realistic scenarios differ, covering
trained vs. untrained parameters and architecture properties. We then move to federated learning with multiple steps
and multiple images, and discuss how to attack this scenario.
Regarding the recovery of batch results, where we show that recovery from a batch of 100 averaged gradients is possible,
the key factor here are the privacy implications. Assuming this was a batch of 100 private photos, would we consider it
secure if only 5 of 100 private photos were revealed? The surprising revelation of the 100 image experiment is that
the distortions arising from batching are not uniform. One could have expected all images to be equally distorted and
near-irrecoverable, yet some images are highly distorted and other only to an extend at which the pictured object can
still be recognized easily. This non-uniformity is a significant result for the privacy of gradient batches. Also note that
Fig.4 of [35] looks better because the attack scenario there is easier. We analyze the attack of [35] in Fig.2 and Tab.1
and find that it struggles in realistic settings.
**Reviewer 3:** We did conduct experiments on fully-connected networks as a sanity check, reaching a full reconstruction.
However, as these merely confirm the proven statement of Prop. 3.1., we did not deem them interesting enough to
include.
Our statement about magnitude and direction of gradients relies on intuitions from optimization, i.e., that the negative
gradient is the direction of steepest descent, and that for strongly convex functions the gradient magnitude is an upper
bound on distance to the optimal solution.
After considering the single-image scenario as in previous works, we do cover more practical scenarios in Sec. 6. The
scenario with multiple participants is equivalent to what we discuss, if the server receives contributions from each
participant separately. If the contributions are averaged without knowledge of the server (such as in secure aggregation),
then recovery of images from multiple participants reduces to recovery from a batch of averaged gradients.
The attack on averaged gradients only has knowledge about the average of gradients, however we assume the number
of participating images to be known to the server. The server might request this information anyway (for example to
balance heterogeneous data), but even if the exact number of images is unknown, the server (which we assume to have
significantly more compute power than the user) could run reconstructions over a range of candidate numbers, given
that the number of images is only a small integer value and then select the solution with minimal reconstruction loss.
We will include more information about computational costs in the revised version of this work. An analysis of attacks
against differentially private models is highly interesting and a topic of future work for us. The reconstruction on
CIFAR for the untrained "LeNet(Zhu)" model is better with L-BFGS because this model is smooth with a large linear
layer. This is a less realistic scenario, but ideal for L-BFGS, and the superior convergence speed of L-BFGS is realized.
Finally, we're glad that you found the code example working as desired.

**Reviewer 4:** We thank you for valuing this work and sharing our enthusiasm in it.

[Meta-Review · NeurIPS 2020]

After discussion and consideration of the author feedback, all reviewers agreed that this paper deserves to be accepted, particularly given the impressive empirical results. Authors, please be sure to include the improvements promised in the feedback (namely, adding visual results corresponding to Table 1, and giving some insight on computational cost).